# Diurnal Cortisol Rhythm in Female Flight Attendants

**DOI:** 10.3390/ijerph18168395

**Published:** 2021-08-08

**Authors:** Małgorzata Radowicka, Bronisława Pietrzak, Mirosław Wielgoś

**Affiliations:** Department of Obstetrics and Gynecology, Medical University of Warsaw, 02-015 Warsaw, Poland; bronislawa.pietrzak@wum.edu.pl (B.P.); miroslaw.wielgos@wum.edu.pl (M.W.)

**Keywords:** cortisol, shift work, flight attendants, cortisol curve

## Abstract

The work of flight attendants is associated with exposure to long-term stress, which may cause increased secretion of cortisol. The aim of the study is to determine the circadian rhythm of cortisol and to seek factors of potential influence on the secretion of cortisol in female flight attendants working within one time zone as well as on long-distance flights. The prospective study covers 103 women aged 23–46. The study group (I) was divided into two subgroups: group Ia, comprising female flight attendants flying within one flight zone, and group Ib, comprising female flight attendants working on long-distance flights. The control group (II) are women of reproductive age who sought medical assistance due to marital infertility in whom the male factor was found to be responsible for problems with conception in the course of the diagnostic process. The assessment included: age, BMI, menstrual cycle regularity, the length of service, the frequency of flying, diurnal profile of the secretion of cortisol, testosterone, estradiol, 17-OH progesterone, SHBG, androstenedione, and progesterone concentration. Descriptive methods and inferential statistics methods were used to compile the data. Comparing the profile of flight attendants from groups Ia and Ib shows that the curve flattened among women flying within one time zone. The secretion curve is also more flattened in women with less years worked and in flight attendants working less than 60 h per month. Due to the character of work, the female flights attendants do not have hypersecretion of cortisol. Frequency of flying and length of work affect the dysregulation of HPA axis.

## 1. Introduction

The human circadian rhythm determines the cyclical changes that take place in the body within 24 h [1]. Sunlight and artificial light synchronize biological rhythms, while the strongest endogenous synchronizer of these rhythms is the suprachiasmatic nucleus (SCN). The genes responsible for the functioning of the biological clock have been found in almost every cell of the body, and the disturbance of its functioning increases the incidence and intensification of the course of many diseases [2]. The presence of light at night and frequent changes in the circadian cycle disrupt the synchronization of rhythms in the body and lead to disturbances in physiological functions. Epidemiological studies show that shift workers are more likely than daytime workers to be exposed to diseases of the cardiovascular, digestive, and reproductive systems, as well as metabolic and hormonal disorders, overweight and obesity, and malignant cancer [3].

The hormones that follow the circadian rhythm are those whose production is regulated by the hypothalamic–pituitary axis. These include cortisol, growth hormone, prolactin, thyroid hormones, and sex steroids [4]. Cortisol is an important hormone regulating the functioning of the cardiovascular and immune systems, as well as responsible for proper metabolism and homeostasis. Cortisol levels rise sharply upon waking in the early morning, then decline during the day, and are lowest during early sleep [5]. The hypothalamic–pituitary–adrenal axis responds to a number of factors, including stress, leading to a temporary increase in cortisol levels [6].

The work of flight attendants is associated with exposure to long-term stress. Research clearly shows that the most stressful part of the flight is landing, especially just before touchdown [7]. Long-term stress may cause increased secretion of cortisol in this occupational group. The aim of the study is to determine the circadian rhythm of cortisol and to seek factors of potential influence on the secretion of cortisol in female flight attendants working within one time zone (more take-offs and more landings) as well as on long-distance flights. This is the first research on cortisol among female flight attendants.

## 2. Material and Methods

### 2.1. Study Design

The cross-sectional study covered 103 Polish women aged 23–46. The women were qualified in the Department of Obstetrics and Gynaecology of the Medical University of Warsaw and examined in the Department of Gynaecological Endocrinology of the Medical University of Warsaw, Poland, in the years 2013–2016. Approval for this study was obtained from the Medical University of Warsaw, Ethics Board (KB/254/2013, 12 November 2013). Notification about the study to be conducted was also given to the trade unions of LOT Polish Airlines and EUROLOT Polish Airlines. Exclusion criteria included the use of hormonal drugs (including contraceptives) up to 6 months prior to the study, use of drugs inducing the activity of hepatic enzymes which can affect the hormonal economy of the organism, history of chronic renal insufficiency and liver cirrhosis, and diagnosis of menopause in women over 40, according to WHO criteria. The trade unions include over five hundred flight attendants. Seventy-four flight attendants responded to the letter of invitation, and forty-three of them qualified for the study, creating study group (I). All patients gave their written informed consent for study participation. Qualifying for the study were female flight attendants who consented to take part in the study, who were of reproductive age, and who worked under the night shift system. The study group (*n* = 43) was divided into two subgroups: Group Ia (*n* = 17), comprising female flight attendants flying within one time zone, and Group Ib (*n* = 26) female flight attendants, working on long-distance flights. The control group (II) comprized 60 women of reproductive age who sought medical assistance due to marital infertility in whom the male factor was found to be responsible for problems with conception in the course of the diagnostic process. Those women work full time (160 h per month) in the public sector as clerks. They do not do night shift work and do not report being exposed to excessive work-related stress.

### 2.2. Sample Collection and Analysis

The examination was performed in accordance with the protocol for diagnosis of hormonal disorders at the Department of Gynaecological Endocrinology, Medical University of Warsaw. The blood tests were carried out twice: between the 4th–6th day of the cycle (the first phase of the cycle) and 2–4 days before the expected menstrual bleeding (the second phase of cycle). The patients were not examined directly after night shifts. Every woman came for the examination after a good night’s sleep. The cortisol, testosterone, estradiol, 17-OH progesterone, androstenedione, and progesterone concentrations in a blood sample collected from the antecubital vein were assessed with an automated enzyme immunoassay. The assessment included age, the body mass index (BMI), menstrual cycle regularity, the length of service (in years), the frequency of flying, cortisol concentration in the first phase of the cycle in the control group, diurnal profile of the secretion of cortisol in the first phase of the cycle in the study group, testosterone, estradiol, 17-OH progesterone, SHBG, androstenedione in the first phase of the cycle, and progesterone concentration in the second phase of the cycle. The assessment did not include smoking, alcohol consumption, physical activity, or dietary habits. The study design is presented in Figure 1.

Menstrual cycles of 25–35 days were considered regular. The frequency of flying was expressed in terms of the number of flying hours per month. To assess the influence of the frequency of flying on the development of hormonal disturbances, the study group of the female flight attendants was divided into two subgroups: women working less than 60 h per month and women working 60 or more hours per month. To assess the influence of the length of work on the development of hormonal disturbances, the study group of the female flight attendants was divided into two subgroups: women who had worked less than 15 years and women who had worked 15 or more years.

### 2.3. Statistical Analysis

Descriptive methods and inferential statistical methods were used to compile the data. The randomness of the study sample was examined in terms of the age and length of work of the patients. To this end, a runs test was applied to check the hypothesis that the way in which patients were selected could be deemed random. Knowing the order in which the patients registered for the study, the randomness of the sample in terms of age and length of work was confirmed. For the qualitative features, the following characteristics were calculated: arithmetic mean (x), median (Me), standard deviation (SD), and coefficient of variation (v%). The chi-square independence test was applied to compare the frequency of the occurrence of individual varieties of features in the study groups and in the subgroups. Where the expected numbers were lower than 5, Yates correction was used in the calculation of the value of the chi-square test. Prior to the comparison of mean values in the study groups and subgroups, the conformity of the distributions of the analyzed measurable variables with the normal distribution was checked with the help of the Shapiro–Wilk test. As the distributions of the majority of the analyzed variables differed significantly from the normal distribution, the comparison of the mean values was chosen to be made with the help of the non-parametric test, rather than the parametric test. As the samples were independent, the Mann–Whitney test was used to compare the mean values. To study the correlation between the measurable variables, the rank correlation coefficient was applied due to distributions significantly different from the normal distribution. The differences between mean values (or frequencies) were found statistically significant when the calculated value of a relevant test was equal or higher than the critical value from relevant tables, with an adequate number of degrees of freedom and the probability of error *p* < 0.05.

## 3. Results

### 3.1. Characteristic of the Study Group and the Control Group

Table 1 shows the characteristics of the study group and the control group, including the subgroups of women flying within one time zone and women serving long-distance flights. 

The mean age of the respondents in the control group (II) was 34.0 SD 4.09 years. Women from the study group (I) were aged 25–43, and the average age was 34.7 SD 4.41 years. 

The length of work (seniority) in the study group ranged from 6 to 25 years. On average, women worked 14.3 SD 5.06 years. Half of the women studied had worked 15 or more years. The women from the study group spent 47 to 85 h per month flying, on average 64.6 SD 7.89 h, while half of them spent flying 65 or more hours per month flying.

### 3.2. Cortisol Secretion

The daily cortisol secretion profile of flight attendants indicates that the mean concentration at 06:00 was 14.37 ± 14.7 µg/dL; at 08:00, 14.35 ± 22.2 µg/dL; at 21:00, 4.07 ± 5.36 µg/dL; and at 22:00, 3.41 ± 6.62 µg/dL. The range of cortisol concentrations at 06:00 was between 3.41 and 103.3 µg/dL; at 08:00, from 2.77 to 144.1 µg/dL; at 21:00, from 1 14 to 33.6 mg/dL; and 22:00, from 1.01 to 42.2 mg/dL.

In group Ia, the mean cortisol concentration at 06:00 was 14.05 ± 6.55 µg/dL, at 08:00, 13.71 ± 10.1 µg/dL; at 21:00, 5.59 ± 7.53 µg/dL; and at 22:00, 5.30 ± 10.1 ng/mL. In group Ib, the daily profile of cortisol secretion was as follows: at 06:00, 14.6 ± 18.6 µg/dL; at 08:00, 14.78 ± 27.7 µg/dL; at 21:00, 3.00 ± 2.63 µg/dL; and at 22:00, 2.15 ± 1.99 µg/dL. The difference in the mean cortisol concentration in the morning in groups Ia and Ib was not statistically significant (*p* > 0.05), while at 21:00, the difference was close to statistical significance (*p* = 0.053), and at 22:00, the mean concentration cortisol was significantly higher in group Ia (*p* = 0.0115). Curves of daily cortisol secretion in the study group and in groups Ia and Ib are presented in Figure 2.

The mean cortisol concentration in the control group was 11.42 ± 3.51 µg/dL, while in the study group it was 14.37 ± 14.7 µg/dL; the difference was not statistically significant (*p* = 0.345), and the cortisol levels were within the norm (Figure 3).

The relationship between the mean excretion of cortisol at 6:00 and the length of service in the study group as well as in groups Ia and Ib was compared. There was no significant correlation in the study group, for which the rank correlation coefficient was close to zero. In group Ia, there was a strong negative correlation between work seniority (in years) and cortisol secretion, i.e., the longer the work experience, the lower the cortisol secretion. On the other hand, in group Ib, flight attendants with longer work experience were characterized by greater secretion of cortisol, but this relationship was not statistically significant. By analyzing the influence of seniority on cortisol secretion, the daily cortisol profile was also compared (Figure 4).

It was shown that in the mornings, the mean secretion of this hormone is lower in patients with work experience up to 15 years, and higher in those working longer, but the difference was not significant (*p* > 0.05). On the other hand, in the evening hours, in the group of flight attendants with shorter working time (seniority), the secretion of cortisol was higher than in the group of flight attendants working longer. The difference in the means in these subgroups at both 21.00 and 22.00 was not statistically significant.

The relationship between the mean cortisol secretion at 6.00 and the number of hours spent in the air in the study group and in subgroups Ia and Ib was examined. In the entire study group, no significant difference was found in the mean cortisol secretion in patients spending up to 60 h in the air per month and in those spending more than 60 h in the air per month (14.05 ± 6.55 µg/dL vs. 14.6 ± 18.6 µg/dL; *p* > 0.05). Additionally, in groups Ia and Ib, the rank correlation coefficients were very close to zero. By analyzing the influence of flying frequency on cortisol secretion, the diurnal cortisol profile was also compared (Figure 5).

It has been shown that in the morning hours, the mean secretion of this hormone is lower in patients flying up to 60 h per month, and higher in those flying more, but the difference was not significant. On the other hand, in the evening hours, in the group of flight attendants flying up to 60 h per month, the secretion of cortisol was higher than in the group of flight attendants flying over 60 h per month. The difference in the means in these subgroups at 21:00 was not statistically significant, but at 22:00, the comparison of the means showed statistical significance (*p* = 0.0115).

### 3.3. Testosterone Concentration

The mean testosterone concentration in the control group was 0.404 ± 0.133 ng/mL. The concentration of this hormone ranged from 0.15 to 0.72 ng/mL. In the study group, the testosterone concentration range oscillated between 0.14 and 0.49 ng/mL and averaged 0.323 ± 0.087 ng/mL. There were significant differences in the mean testosterone concentration between the study group and the control group (*p* < 0.01), as well as between Ia and Ib (0.283 ng/mL vs. 0.348 ng/mL; *p* < 0.05). The relationship between testosterone secretion and the number of hours spent in the air was investigated, and it was found that the mean testosterone concentration was higher in women who fly more frequently (0.348 ng/mL vs. 0.283 ng/mL; *p* < 0.05). When analyzing the relationship between testosterone secretion and work experience, no statistically significant difference was found.

### 3.4. Progesterone Concentration

The mean concentration of progesterone in the control group was 10.2 ± 4.91 ng/mL. The concentration of this hormone ranged from 1.33 to 23.3 ng/mL. In the study group, the range of progesterone concentrations oscillated between 0.29 and 16.89 ng/mL and averaged 5.14 ± 3.84 ng/mL. There were significant differences in the concentration of progesterone between the study and control groups, as well as between the Ia and control groups (5.93 ng/mL vs. 10.3 ng/mL; *p* < 0.01) and Ib and control groups (5.09 ng/mL vs. 10.3 ng/mL; *p* < 0.001). Progesterone concentration > 10 ng/mL in the second phase of the cycle, indicating ovulation, was found in 6 flight attendants (15.4%) and in 30 women from the control group (50%). There was a significant difference between the study group and the control group (*p* < 0.001); the results of progesterone concentration above 10 ng/mL were more often observed in the control group than in the study group—50.0% vs. 15.4%. When assessing the secretion of progesterone, no significant relationship was found between the average concentration of this hormone among women working over 15 years and those working under 15 years. On the basis of comparing the relationship between the mean concentration of progesterone and the frequency of flying, no significant difference was found. Progesterone concentrations in patients working in the air for up to 60 h per month, compared to those working more than 60 h, were similar: 5.21 ± 4.42 ng/mL and 5.09 ± 3.28 ng/mL.

### 3.5. 17-Hydroxyprogesterone Concentration

In the control group, the mean concentration of 17-OHP was 0.888 ± 0.47 ng/mL. The concentration range was 0.19–3.36 ng/mL. In the study group, the concentration range of 17-OHP oscillated between 0.20 and 2.15 ng/mL and averaged 0.671 ± 0.46 ng/mL. There were significant differences in the concentration of 17-OHP between the study group and the control group (*p* < 0.01), as well as between the Ia group and the control group (0.579 ng/mL vs. 0.888 ng/mL; *p* < 0.05). The relationship between 17-OHP secretion and the number of hours spent in the air was investigated. There was no statistically significant difference, but the mean concentration in the group of patients flying more frequently was much higher (0.723 ± 0.562 ng/mL vs. 0.579 ± 0.233 ng/mL; *p* = 0.877). When analyzing the relationship between 17-OHP secretion and work experience, no statistically significant difference was found. In flight attendants who had worked less than 15 years, the mean concentration of 17-OHP was higher than in the group of patients with longer work experience (0.771 ± 0.60 ng/mL vs. 0.603 ± 0.34 ng/mL; *p* = 0.440).

Table 2 shows the level of estradiol, androstenedione, and SHBG concentration in the study group and the control group. No statistically significant difference was found between the groups studied (I, II, Ia, Ib).

## 4. Discussion

Stress is a factor that stimulates the hypothalamic–pituitary–adrenal axis (HPA). Frequent exposure to stressors causes repeated activation of HPA, disrupting it, leading to unfavorable health consequences [8]. Cortisol is mainly produced by the adrenal glands and is used as a biomarker of HPA function. For most people, when they wake up, their cortisol levels rise immediately, peaking around 20–30 min later, then dropping for the rest of the day [5]. The flattening of the circadian cortisol profile is believed to reflect dysregulation of the hypothalamic–pituitary–adrenal axis [9].

When examining the daily profile of cortisol in flight attendants, we assessed its concentration at 6:00, 8:00, 21:00, and 22:00. Comparing the profile of flight attendants from groups Ia and Ib shows that the curve flattened among women flying within one time zone. In this group, flight attendants experience more take-offs and landings, which are the most stressful parts of the flight. They are also more often exposed to changes in atmospheric pressure. MacDonald et al. assessed the work-related stress of flight attendants and stated that, among others, flight attendants flying on domestic flights show greater fatigue and tension related to work than flight attendants flying long-haul flights [10]. To our knowledge, the effect of changing atmospheric pressure on blood cortisol levels has not been studied so far. In the study, we also assessed the effect of seniority (years worked) and frequency of flying on the daily cortisol profile. The secretion curve was more flattened in women with less work experience (years worked) and in flight attendants working less than 60 h per month. It is assumed that greater work experience (longer work experience, more frequent flying) reduces the stress of flying. Studies have shown that the presence of a flattened curve of circadian cortisol secretion positively correlates with post-traumatic stress disorder, depression, chronic fatigue syndrome, and deaths due to cardiovascular diseases [11].

Diez et al. assessed, among other things, cortisol levels in saliva in 47 bus drivers. The study group was divided into drivers working on morning shifts (early shift work) and afternoon shifts. A flattening of the cortisol secretion curve was obtained in the group of drivers working on the morning shift, which was explained by a much shorter sleep time in those drivers than in drivers working on the afternoon shift [12]. Bostock et al. obtained similar results when studying 30 male pilots flying on short-haul flights. They compared the diurnal profile of cortisol during early and late shift, finding that a much slower decrease in cortisol occurs when pilots work on morning shift [13]. The above studies suggest that shift work, to which flight attendants are also exposed, is a factor that disrupts the hypothalamic–pituitary–adrenal axis. 

Progesterone is a precursor in the biosynthesis of cortisol [14]. The mean concentration of progesterone in the study group (5.44 ng/mL) was significantly lower than in the control group. Progesterone concentration above 10 ng/mL in the second phase of the cycle, indicating ovulation, was found only in six flight attendants (15.4%). The antagonistic effect of the hypothalamic–pituitary–adrenal axis on the hypothalamic–pituitary–gonadal axis may explain the low concentration of progesterone in the study group. Glucocorticosteroids inhibit HPG by initiating the synthesis of gonadotropin-inhibiting hormone (GnIH) at the expense of gonadotropin-releasing hormone (GnRH) [15]. 

Steroid hormones such as androgens are also subject to the circadian rhythm, and their concentration fluctuates with the sleep-wake phase [16]. Due to the nature of the work, flight attendants are prone to sleep disturbance. In the study, we showed that both testosterone and 17-hydroxyprogesterone levels were significantly lower than in the control group. Slow-wave sleep plays an important role in regulating the synthesis and secretion of androgens [16]. Ukraintseva et al. investigated the effect of suppression of slow-wave sleep (SWS) on, among others, morning androgen levels and demonstrated reduced testosterone and 17-hydroxyprogesterone levels in men subjected to SWS suppression [16]. The results of this study suggest that chronic sleep problems, especially those that interfere with deep sleep, increase the risk of developing androgen deficiency. The exact mechanisms involved in the effects of SWS suppression on steroidogenesis are unknown, so it is unclear whether hormone secretion is impaired by sleep deprivation itself or by a deficiency in certain specific sleep phases critical to steroid release. Taking into account the influence of androgens on carbohydrate metabolism, cognitive functions, and mood, it can be concluded that changes in androgen secretion may significantly influence the development of metabolic and psychiatric disorders.

The strengths of this study include the fact that it is the first research on cortisol among female flight attendants. Cortisol concentration, expressed as a diurnal profile of the secretion, further enhances the accuracy of the presented research. The study also compares cortisol secretion with other hormones, which has been less investigated so far. We recognize a limitation in this study. We did not collect data on glucose profile and on hemodynamic parameters such as heart rate variability and blood pressure profile, which are associated with cortisol secretion. We also did not collect data on chronic stress in the control group. Our studied group is small, so it was difficult to find significant relationships from the data, as statistical tests normally require a larger group to ensure a representative distribution of the population and to be considered representative of groups of people to whom results will be generalized or transferred.

## 5. Conclusions

Due to the character of work, female flight attendants do not have hypersecretion of cortisol.Frequency of flying and length of work affect the dysregulation of the HPA axis.Greater work experience (longer work experience, more frequent flying) reduces the stress of flying.Further research on a larger cohort of flight attendants is necessary to assess metabolic and hemodynamic parameters that may affect the cortisol profile.

## Figures and Tables

**Figure 1 ijerph-18-08395-f001:**
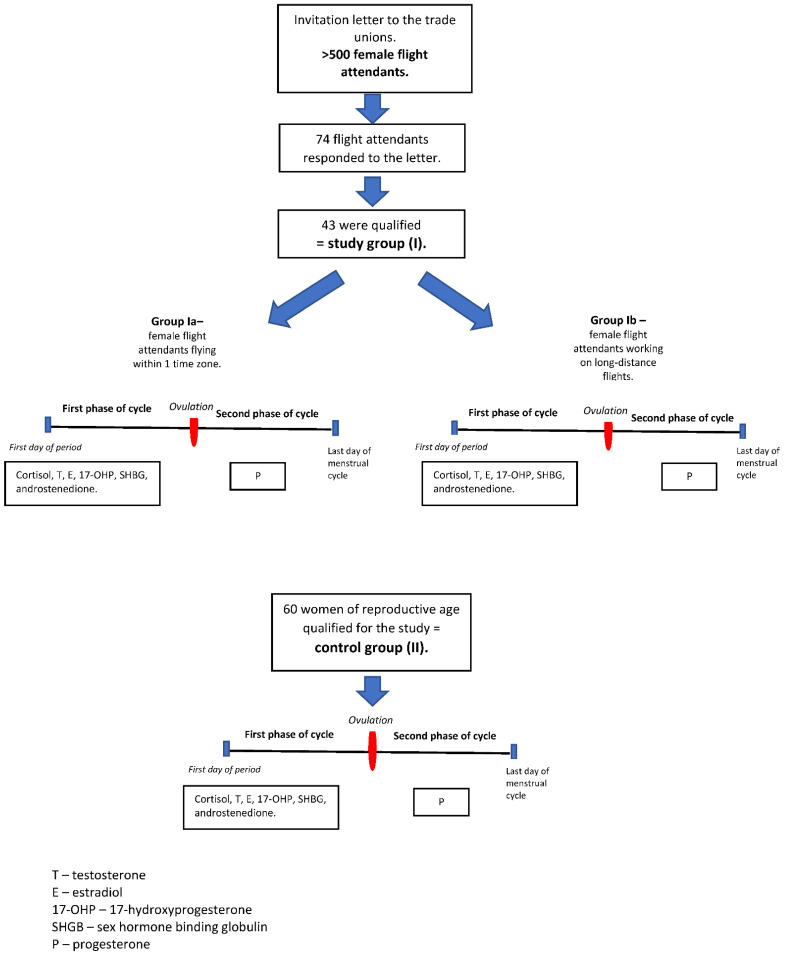
Study design.

**Figure 2 ijerph-18-08395-f002:**
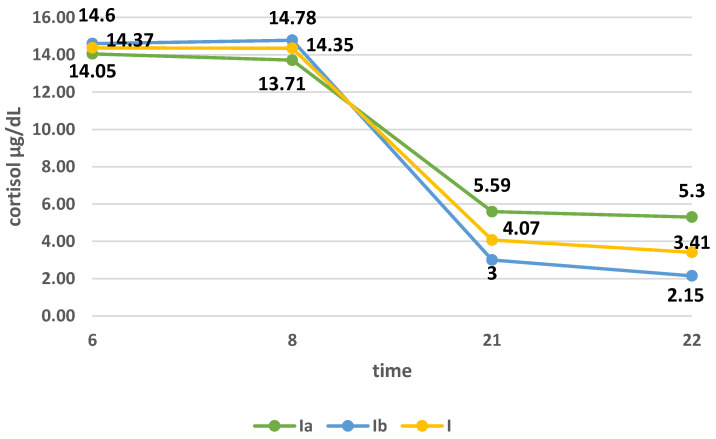
The diurnal profile of the secretion of cortisol in the study group and in groups Ia and Ib.

**Figure 3 ijerph-18-08395-f003:**
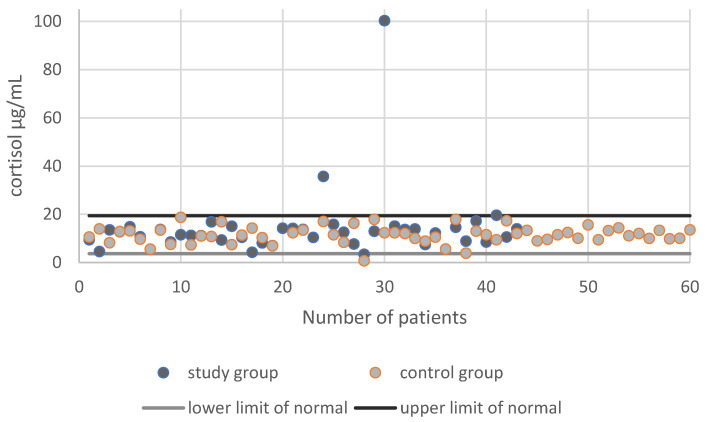
The cortisol concentration in the study group and in the control group.

**Figure 4 ijerph-18-08395-f004:**
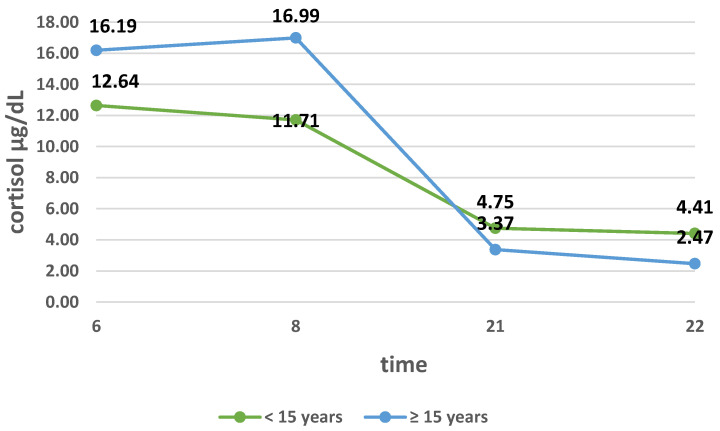
Diurnal cortisol profile depending on seniority.

**Figure 5 ijerph-18-08395-f005:**
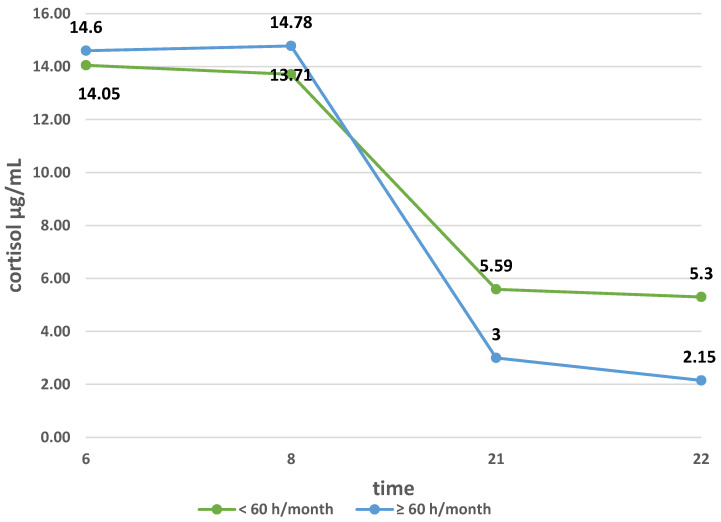
Diurnal cortisol profile depending on the frequency of flying.

**Table 1 ijerph-18-08395-t001:** The characteristics of the study group and control group, including the subgroups of women flying within one time zone and women serving long-distance flights. Time of work in air was significantly different between groups Ia and Ib.

Variable	Control Group (II) (*n* = 60)			Study Group (I) (*n* = 43)				Group Ia (*n* = 17)			Group Ib (*n* = 26)			
	Min	Max	M ± SD	Min	Max	M ± SD	*p*	Min	Max	M ± SD	Min	Max	M ± SD	*p*
Age (years)	23	46	34.0 ± 4.09	25	43	34.7 ± 4.41	0.203	26	43	34.1 ± 4.26	25	40	35.2 ± 4.53	0.219
BMI (kg/m^2^)	17.5	37	22.7 ± 3.78	17.5	29.0	22.4 ± 2.71	0.995	17.5	28	21.7 ± 2.67	19	29	22.8 ± 2.71	0.28
Seniority (years)				6	25	14.3 ± 5.06		6	18	13.0 ± 4.23	6	25	15.2 ± 5.44	0.084
Time of work in air (h/month)				47	85	64.6 ± 7.89		47	70	59.5 ± 6.80	55	85	67.9 ± 6.81	**0.001**

Note: Bolded is *p* < 0.001; Control group (II)—60 women of reproductive age who sought medical assistance due to marital infertility in whom the male factor was found to be responsible for problems with conception in the course of the diagnostic process; Study group (I)—43 female flight attendants; Group Ia—17 female flight attendants flying within one time zone; Group Ib—26 female flight attendants working on long-distance flights. Bold: mark significant result.

**Table 2 ijerph-18-08395-t002:** The concentrations of the estradiol, SHBG, and androstenedione in individual groups.

	Control Group (II) (*n* = 60)			Study Group (I) (*n* = 43)				Group Ia (*n* = 17)			Group Ib (*n* = 26)			
	Min	Max	M ± SD	Min	Max	M ± SD	*p*	Min	Max	M ± SD	Min	Max	M ± SD	*p*
Estradiol (pg/mL)	10	303	47 ± 39	9	139	48.8 ± 26.9	0.397	9	107	44.6 ± 23.4	19	139	51.6 ± 29.1	0.504
SHBG (nmol/L)	24.6	175.7	72 ± 36.1	38.1	164.9	71.35 ± 25.8	0.491	38.1	122.2	65.87 ± 23.7	41.3	164.9	74.64 ± 27	0.154
Androstenedione (ng/mL)	1	6	2.89 ± 1.11	1.4	9.4	2.97 ± 1.83	0.553	1.4	4.8	2.71 ± 1.34	1.7	9.4	3.16 ± 2.16	0.591

Control group (II)—60 women of reproductive age who sought medical assistance due to marital infertility in whom the male factor was found to be responsible for problems with conception in the course of the diagnostic process; Study group (I)—43 female flight attendants; Group Ia—17 female flight attendants flying within one time zone; Group Ib—26 female flight attendants working on long-distance flights.

## Data Availability

The data presented in this study are available on request from corresponding author. The data are not publicly available due to polish law regulation does not allow for public access of medical data.

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
