# Peer review of "Diurnal Cortisol Rhythm in Female Flight Attendants"

_ijerph, 2021, doi:10.3390/ijerph18168395_

Round 1

Reviewer 1 Report

This study was examined the daily profile of cortisol in flight attendants. The results showed that the secretion curve was more flattened in women with less work experience (years worked) and in-flight attendants working less than 60 hours per month. Concluding that greater work experience (longer work experience, more frequent flying) reduces the stress of flying. This study addresses a theme appropriate to the journal. The study reports interesting results and is well structured, but with some points to be clarified and reviewed.

Minor issues:

  • Headings must not be present in the abstract, remove them.
  • Line 86: The study group was divided into two subgroups: Group Ia (n = 17) comprising of female flight attendants flying within one time zone and Group Ib (n = 26) female flight attendants working on long-distance flights. The previous paragraph, and the abstract, report: The cross-sectional study covered 103 Polish women. I don't understand exactly how many people were included in the study.
  • Improve image 1, both graphically and conceptually.
  • Improve table 1, adding rows and columns, be more precise.
  • In Figures 2,3 and 4, replace the axis title, cortisol concentration µg / dl with cortisol [µg / dl], and hour with time.

Reviewer 2 Report

Dear Authors,

Thank you for submitting this paper. I enclose some suggestions to improve your manuscript:

ABSTRACT – In the abstract (as well as in the introduction), I would not use the past tense.

INTRODUCTION – I would clearly state whether (and why) your contribution is original with respect to other works on the topic.

MATERIALS AND METHODS – A literature review section is missing. Please include a paragraph with recent literature on the topic.

RESULTS AND DISCUSSION – Findings are well presented. In Discussion you should interpret and describe the significance of your findings in light of what was already known about the topic (the theoretical framework, that you should include in a dedicated paragraph inside Section 2).

STRUCTURE OF THE PAPER – I would include a “Conclusions” section, where to clearly answer to the research questions stated in the introduction, as well as to highlight possible limitations of the study, its practical implications and directions for future research.

Reviewer 3 Report

1.The Study Design section is very confusing. If you are going to explain according to Figure 1, you should provide the references in the text before explaining.

2.Regarding the sample collection, is there any variation depending on the collection time? How much of a time gap was considered acceptable?

3.How was the sample size calculated and set this time?

Reviewer 4 Report

In their work, Radowicka et al. present an interesting perspective on diurnal cortisol rhythm in female flight attendants stratified into those that work within a single time zone and those that work in the setting of long flights. The control group to which they compare hormonal dynamics was comprised of women of reproductive age who sought medical assistance due to marital infertility in whom the male factor was found to be responsible for problems with conception. They report that the daily secretion curves of cortisol were similar across groups, however, those flight attendants that worked longer and higher work experience had significantly lower cortisol secretion. They also report that frequency of flying and length of work affect the dysregulation of the HPA axis.

All ethical considerations and approvals have been appropriately disclosed.

There are some comments:

  1. In Figures 2, 4, 5 please use more distinct colors for the lines (e.g. blue, red, etc.) because for example in Figure 2 three different shades of grey are presented which might be difficult to follow.
  2. The results section in the present form is difficult to read and follow, it is way too condensed. I would strongly recommend the authors to make a breakdown of this section with respect to paragraphs and adding subsections (with subtitles accordingly).
  3. It would be interesting to see if stress with respect to cortisol secretion could be linked to parameters of metabolism such as potential glucose profile dysregulation and other parameters such as heart rate, etc.
    Please consider incorporating this aspect in the discussion, for example, Bozic et al. reported in their work on morning cortisol levels in patients with obstructive sleep apnea that these levels were strongly linked to glucose metabolism parameters. Please consider expanding your work with respect to this aspect and include references such as Bozic et al. Endocrine. 2016;53:730-9.
  4. You should list some of the limitations of your work in the separate paragraph of the Discussion section such as the retrospective nature of the study, etc.
  5. I think that it would be worthwhile that autonomic and basic hemodynamic parameters such as heart rate, heart rate variability, blood pressure profile, etc. could be added and examined with respect to these hormonal parameters. If these were not measured (as it is most likely), this should be acknowledged as a limitation in this work.

Round 2

Reviewer 2 Report

Dear Authors,

Thank you for the revised version of the paper, which has consistently improved with respect to the original one.

Best wishes and congratulations